# Ketols Emerge as Potent Oxylipin Signals Regulating Diverse Physiological Processes in Plants

**DOI:** 10.3390/plants12112088

**Published:** 2023-05-24

**Authors:** Katherine M. Berg-Falloure, Michael V. Kolomiets

**Affiliations:** Department of Plant Pathology and Microbiology, Texas A&M University, College Station, TX 77843, USA; kathermb@tamu.edu

**Keywords:** abiotic stress, conjugation to catecholamines, CYP74 enzymes, induced systemic resistance, induction of flowering, ketol, systemic acquired resistance, plant–insect interactions, plant–pathogen interactions

## Abstract

Plants produce an array of oxylipins implicated in defense responses against various stresses, with about 600 oxylipins identified in plants to date. Most known oxylipins are the products of lipoxygenase (LOX)-mediated oxygenation of polyunsaturated fatty acids. One of the most well-characterized oxylipins produced by plants is the hormone jasmonic acid (JA); however, the function of the vast majority of oxylipins remains a mystery. One of the lesser-studied groups of oxylipins is comprised of ketols produced by the sequential action of LOX, allene oxide synthase (AOS), followed by non-enzymatic hydrolysis. For decades, ketols were mostly considered mere by-products of JA biosynthesis. Recent accumulating evidence suggests that ketols exhibit hormone-like signaling activities in the regulation of diverse physiological processes, including flowering, germination, plant–symbiont interactions, and defense against biotic and abiotic stresses. To complement multiple reviews on jasmonate and overall oxylipin biology, this review focuses specifically on advancing our understanding of ketol biosynthesis, occurrence, and proposed functions in diverse physiological processes.

## 1. The Lipoxygenase Pathway: CYP74 Enzymes

Plants produce a large number of diverse oxygenated lipids, collectively called oxylipins, that play a role in all aspects of plant physiology, including growth, development, and defense [1,2,3,4]. Lipoxygenases (LOX) oxidize polyunsaturated fatty acid (PUFA) substrates and are the first enzymatic step for the synthesis of most oxylipins [5]. In plants, the major PUFA substrates are linoleic (C18:2), linolenic (C18:3), and hexadecatrienoic acid (C16:3). Depending on the oxidized carbon position of C18:2 or C18:3, LOXs produce either 9- or 13-hydroperoxides of these fatty acids, or 11- or 7-hydroperoxides of the C16:3 substrate. Fatty acid hydroperoxides are fluxed into seven subsequent branches of the LOX pathway, which collectively synthesize an array of structurally and functionally diverse oxylipins [6]. These LOX pathway branches include allene oxide synthase (AOS), hydroperoxide lyase (HPL), divinyl ether synthase (DES), epoxyalcohol synthase (EAS), reductase (RED), peroxidase (POX), and lipoxygenase (LOX) [5]. There are a number of excellent reviews that provide detailed overviews of several of these branches and their oxylipin products [6,7,8,9,10,11,12].

This review focuses on the synthesis and functions of ketol oxylipins produced by enzymatic actions of the AOS members of the CYP74 subfamily of enzymes (Figure 1). AOS, HPL, EAS, and DES belong to the CYP74 subfamily of cytochrome P450 enzymes, divided into subfamilies designated as CYP74A, B, C, and D based on their specific enzymatic activity [13,14]. CYP74A displays 13-AOS or EAS activity, CYP74B contains enzymes with13-HPL activity, the members of CYP74C contain enzymes with 9/13-HPL and 9/13-AOS activity, and the CYP74D subfamily is composed of enzymes displaying 9/13-DES activity [15].

The 13-HPL pathway leads to the production of C6 volatiles, referred to as green leafy volatiles (GLVs), and C12 compounds, collectively termed traumatins (Figure 1). The 9-HPL pathway produces both volatile and non-volatile C9 compounds. The 9- and 13-DES pathways are responsible for the production of divinyl ethers. The 13-DES activity leads to the production of etheroleic or etherolenic acid, while the 9-DES activity produces colneleic or colnelenic acid [11]. The EAS pathway gives rise to epoxyalcohols [15], with EAS enzymes recently identified in higher plants as part of the CYP74 family [16].

In all plant studied species, the 13-AOS branch of the LOX pathway is known to produce the well-characterized plant hormones, 12-oxo-phytodienoic acid (12-OPDA) and jasmonic acid (JA), and their derivatives collectively called jasmonates. The 12-OPDA is produced by the sequential action of 13-LOX, 13-AOS, and 13-AOC, which are all localized to plastids. Subsequently, 12-OPDA is transported to peroxisome for further conversion to JA. The 13-AOS pathway also produces 13-ketols, including 9-hydroxy-12-oxo-10(E)-octadecenoic acid (9,12-KOMA), 9-hydroxy-12-oxo-10(E),15(Z)-octadecadienoic acid (9,12-KODA), 13-hydroxy-12-oxo-9(Z)-octadecenoic acid (13,12-KOMA), and 13-hydroxy-12-oxo-9(Z),15(Z)-octadecadienoic acid (13,12-KODA) (Figure 2). The 9-AOS pathway produces 10-oxo-11(Z)-phytoenoic acid (10-OPEA), 10-oxo-11(Z),15(Z)-phytodienoic acid (10-OPDA), and their derivatives collectively called “death acids” due to their strong programmed cell death-inducing activity [17]. Recently, an additional AOS branch of oxylipins, collectively named ‘graminoxins’, was identified in wheat, barley, sorghum, and rice roots [18], the function of which is currently unknown. Products from the 9-AOS pathway can undergo spontaneous cyclization into 10-OPEA and 10-OPDA, with speculation that a putative 9-AOC exists in this pathway [17,19]. The allene oxides produced by the 9-AOS pathway are also converted non-enzymatically to 9-ketols, including 9-hydroxy-10-oxo-12(Z)-octadecenoic acid (9,10-KOMA), 9-hydroxy-10-oxo-12(Z),15(Z)-octadecadienoic acid (9,10-KODA), 13-hydroxy-10-oxo-11(E)-octadecenoic acid (13,10-KOMA), and 13-hydroxy-10-oxo-11(E),15(Z)-octadecadienoic acid (13,10-KODA) (Figure 2).

It should be noted that 9-LOX or 13-LOX activity gives rise to structurally diverse metabolites depending on whether these metabolites are derived from C18:2 or C18:3 fatty acids (Figure 1). Many of these oxylipins have antimicrobial and/or antifungal properties and, thus, are suspected of playing important roles in defense against biotic stresses; while others are implicated to have signaling roles in the regulation of plant development and reproduction [5,6,11,20,21]. However, the exact roles of the vast majority of oxylipins remain uncharacterized. To highlight the physiological functions of ketols, this review focuses on our current knowledge of their synthesis, biological activities, and diverse reported functions.

## 2. Biosynthesis and Occurrence of Ketols

Ketols are C18 compounds that contain a hydroxide group (-OH) present at the 9- or 13-carbon of the fatty acid backbone, a ketone group (C=O) present at the 10- or 12-carbon and ending in a carboxyl functional group (Figure 2). Ketols are designated as ‘α-’ or ‘γ-’ ketols based on the location of the hydroxide functional group in relation to the ketone functional group. If two double-bonds are present, then these ketols are referred to under the abbreviation of ‘KODA’ and are produced from C18:3 substrate, while if a single double-bond is present, then the ketols are referred to as ‘KOMA’ and are produced from the C18:2 substrate [22]. Ketols are grouped into ‘9-AOS-derived ketols’ (9-ketols for short) or ’13-AOS-derived ketols’ (13-ketols) based on the regiospecific 9- or 13-AOS enzymes that produce them. It should be noted that the following four ketols, 9,12-KODA, 9,12-KOMA, 13,10-KODA, and 13,10-KOMA, are classified as reactive electrophilic species (RES) because they contain an α,ß unsaturated carbonyl group (Figure 2). α,ß-unsaturated carbonyls are known to interact with nucleic acids and proteins to initiate adverse biological effects [23].

The proposed mechanism for the synthesis of ketols begins with the conversion of 9- or 13-hydroperoxide fatty acid substrates into epoxide intermediates via AOS activity (Figure 3). In this specific example, hydrogen donated from a hydronium (H_3_O^+^) ion attaches to the oxygen in the epoxide ring, leading to the breakage of the epoxide ring and the formation of a hydroxyl functional group on the 10-carbon position (Figure 3). Subsequent hydrolysis reactions occur to add another hydroxyl functional group to the 13-carbon position, leading to the transfer of electrons through double bonds and the formation of an ‘enol’ product that undergoes tautomerization (Figure 3). In the final step of ketol synthesis, a compound acting as a base attracts the hydrogen attached to the carbonyl functional group on the 10-carbon, leading to the ketol product of 13,10-KODA in this specific example (Figure 3).

In addition to 13-AOS that all characterized plant species possess, many monocot species encode an additional clade of putatively extraplastidic AOS isoforms that possess dual 9/13-AOS activity [6,24]. It is likely that these 9/13-AOS isoforms are responsible for the biosynthesis of 9-oxylipins, death acids, and 9-ketols in addition to 13-ketols. Currently, little information is known about the regulation of the synthesis of ketols. It is suggested that ketol production is dependent on JA, as the expression of LOX and AOS genes involved in ketol production is dependent on JA [6], and disruption of JA biosynthesis in maize results in reduced levels of wound-induced ketols [25].

As illustrated in Figure 2, the 13-AOS enzymatic branch occurs primarily in plastids where it acts upon hydroperoxides of either C18:3 or C18:2 acids produced by 13-LOXs [5]. The products of 13-AOS are short-lived epoxide intermediates that are either non-enzymatically hydrolyzed to form α- and γ- ketols or undergo enzymatic cyclization by 13-AOC, the latter pathway leading to the eventual production of JA. Due to C18:3 being the predominant fatty acid found in the glycerolipids present in plastid membranes, a higher percentage of products derived from C18:3 than C18:2 is expected to be produced in these organelles.

Typically, AOSs exhibiting 13-AOS activity are categorized as ‘CYP74A’ enzymes [26]. It is important to note that cyclization of the allene oxide intermediate synthesized under the 13-AOS pathway is possible with the presence of 13-AOC. It has been recently shown that 13-LOXs, 13-AOS, and 13-AOC form a protein complex to channel substrate specifically for the synthesis of JA [27]. However, it is not known if the formation of this complex interferes with ketol synthesis through substrate competition.

The majority of tested 9-LOXs and 9-AOSs have been reported to be localized to the cytosol and/or organelles other than the chloroplast [6,28,29]. To date, all characterized cytosolic AOSs possessing 9-AOS activity display a strong affinity for both 9- and 13-hydroperoxide substrates; thus, they are characterized as dual-specific enzymes [6,19,23,28,29,30,31,32]. Typically, AOSs that have this dual-specific activity are grouped within the CYP74C subfamily [26,28]. Examples of dual-specific AOS genes include a barley enzyme capable of producing α-ketols from both 9- and 13-hydroperoxides of fatty acids [23]. The 9/13-AOSs were identified in other monocot species, including maize [6,19], rice [33], and sugarcane [34], based on their phylogenetic relationship and ability to produce both 9- and 13-ketols. Some dicot species also contain a similar mixed-function AOS, including tulips [35], tomatoes [26], and potatoes [28].

The best-characterized JA-producing and plastid-localized 13-AOS enzyme is present in both dicot and monocot plant species. Plant species are known to contain different numbers of AOS genes ranging from a single 13-AOS gene in *Arabidopsis* to three 13-AOS genes and two 9/13 mixed function AOS genes in maize [6], to twelve mixed function 9/13-AOS genes in sugarcane [34]. The mixed function 9/13-AOSs have been reported in fewer dicot species than monocot species. Flax, barley, maize, rice, tulips, petunia, potato, and tomato are among a few of the plant species identified so far that contain a dual-specific 9/13-AOS enzyme and therefore synthesize both 9- and 13-ketols [6,19,24,28,29,30,31,32]. Little is known about the evolution of CYP74 enzymes, especially AOS genes. Although *Arabidopsis* AOS shares structural similarities to other enzymes in mammals [14], it is not known whether mammalian systems possess AOS-like enzymes. Interestingly, 9/13-AOS genes are present in lancelets, which are considered to be an evolutionary intermediate between vertebrates and invertebrates [36]. This recombinant enzyme from the lancelet, *Branchiostoma belcheri* Gray, was active towards both 9- and 13-hydroperoxides, producing ketols 13,12-KOMA, 13,12-KODA, 9,10-KOMA, and 9,10-KODA [36]. Both soft and stony corals also possess AOS enzymes, though it is not reported if these enzymes produce ketols [37,38]. Because liverworts, mosses, and green algae contain AOS enzymes [29,30,32], it is suggested that the introduction of AOS genes occurred before the evolution of terrestrial plants, and AOS enzymes were likely present in the last common ancestor of plants and animals [14,34]. Recently, a metabolic analysis of wounded *Physcomitrium patens* Mitt knockout mutants of the PpAOS1 gene uncovered that this enzyme is responsible for the synthesis of two α-ketols [39]. Available data suggest that dual-specific AOS enzymes localized either to cytosol or chloroplast likely evolved before the evolution of flowering plant species [14,29,34].

## 3. Ketols Serve Signaling Roles in Plant–Pathogen Interactions

Several studies implicated ketols in playing a role in defense against biotic stresses. Such a role was first suggested for ketols produced in below-ground organs, as these tissues are characterized by especially high activity of 9/13-AOS enzymes and high levels of ketol production [26,28,35]. Recent research suggests that ketols are involved in plant–pathogen interactions in above-ground tissues. Exogenous application of 9,10-KODA results in strong induction of expression of several SA-inducible PR genes in tobacco (*Nicotiana tabacum*) leaves to the extent similar to the effect of the application of SA, suggesting that 9,10-KODA may be associated with defenses against pathogen infections and systemic acquired resistance (SAR) [40]. While this study provided strong evidence for the potential relevance of 9,10-KODA to SAR induction, it is unknown if 9,10-KODA acts independently of SA on the induction of SAR marker genes. A further report supported the defensive role of 9,10-KODA as an exogenous application of 9,10-KODA to grape berries suppressed the disease progression by *Glomerella cingulata* Stonem [41]. Recent genetic and pharmacological evidence obtained by the analysis of maize knockout mutants identified 9,10-KODA as a major xylem-mobile long-distance signal required for the activation of induced systemic resistance (ISR) against leaf pathogens triggered by root colonization with the beneficial fungal symbiont *Trichoderma virens* [42], suggesting a signaling role for this molecule. Moreover, several γ-ketols were subsequently shown as additional ISR priming agents induced transiently in leaves of maize in response to *T. virens*-triggered ISR [22]. Importantly, these studies indicated that both 9- and 13-ketols serve as important signals for the induction of ISR in maize [22,42]. Additionally, several ketols, including 9,10-KODA, 9,12-KODA, 13,12-KODA, and 13,12-KOMA, were upregulated in *Colletotrichum graminicola*-infected maize plants that were treated with pentyl leaf volatiles (PLVs); suggesting a positive correlation with PLV-mediated pathogen resistance [43]. The 9,10-KODA and other 9-LOX products were highly induced upon infection of maize stems by the hemibiotrophic pathogen *Fusarium graminearum* Petch, the causal agent of Gibberella Stalk Rot (GSR) [44]. Significantly higher accumulation of this ketol was observed in the maize line resistant to GSR, whereas *Zmlox5* mutants disrupted in 9,10-KODA production displayed increased susceptibility to GSR [44].

## 4. The Involvement of Ketols in Plant–Herbivore Interactions

The role of ketols in plant–herbivore interactions is vastly understudied. However, there is evidence that these metabolites serve an important function in such interactions, as many studies highlight the upregulation of putative ketol-producing AOS genes in response to herbivore feeding [1,45,46,47]. In maize, ZmAOS2b (which has 9/13-AOS activity and is predicted to be extraplastidic [19]) was upregulated during herbivory by Beet Armyworm (BAW) but was no longer inducible in the herbivory-susceptible JA-deficient *opr7opr8* mutant [1]. This mutant was unable to produce normal levels of both 9- and 13-ketols in response to wounding [25], suggesting that ketol synthesis is JA-dependent. The importance of 9/13-AOS enzymes is evidenced further by the transcriptome study of Tzin et al. [47], which showed rapid and transient induction of ZmAOS2a and ZmAOS2b in response to feeding by BAW [47]. Moreover, oral secretions from fall armyworm (FAW) induce transcript accumulation of *ZmAOS2b*, as maize plants infested with unablated FAW caterpillars induced its expression ~80-fold while feeding by ablated FAW resulted in only ~20-fold induction [45].

Additionally, maize infested with aphids displayed strong induction of both 9/13-AOSs, *ZmAOS2a*, and *ZmAOS2b* at different time points after infestation [46]. ZmAOS2b reached an 8-fold induction at 48 h after feeding, while ZmAOS2a reached a 15-fold induction at 48 h [46]. Since neither of these enzymes is likely to contribute towards JA production [6], this suggests that ketols and/or death acids produced by these enzymes are important for defense against sucking insects.

Recent genetic evidence obtained by the analysis of MpAOS1 and MpAOS2 in the liverwort *Marchantia polymorpha* showed that disruption of MpAOS1 and MpAOS2 displayed increased susceptibility of the plant to the spider mite *Tetranychus urticae* [48]. It was shown that the recombinant proteins of MpAOS1 (localized to cytosol) and MpAOS2 (localized to chloroplasts) displayed strong activity towards the synthesis of α-ketols [48]. The literature also suggests that ketols are important for defense against Root Knot Nematodes (RKN), as several 9- and 13-AOS products such as 10-OPEA, 9,10-KOMA, and 9,12-KOMA were upregulated in tomatoes in response to RKN infection by *Meloidogyne* spp. [49]. Furthermore, *ZmAOS2b* was strongly induced during the course of maize root infestation by RKN [50].

## 5. Ketols Are Involved in Abiotic Stress Response

Accumulating evidence suggests that ketols may possess regulatory activities in certain abiotic stress responses. Gorina et al. [51] found that potato StAOS3, which shares 9-AOS activity similar to LeAOS3 and ZmAOS2b, was upregulated in roots after salinity stress and above-ground tissues following dark stress treatment [51]. The 9,10-KOMA production in root extracts incubated with C18:2 and C18:3 fatty acids was increased after roots were exposed to dark and herbicide stress, with a slight elevation in 9,10-KODA levels [51]. Exogenous application of 9,10-KODA to wheat seeds was also shown to be an important regulator of seed germination during drought stress, root elongation under alkalinity conditions, and overall wheat yield as plants grown from seeds imbibed with this 9-ketol displayed improved performance [52].

Interestingly, plants are not the only organisms that display upregulation of AOS genes during abiotic stress. The AOS-LOX gene of the soft coral *Capnella imbricata* was upregulated during heat stress [53]. This gene was most highly induced at a lower (28 °C) heat stress condition rather than at a higher (31 °C) temperature, implying that the AOS-LOX gene in soft coral may be important in the early detection of heat stress [53].

## 6. Ketols Contribute to the Normal Growth and Reproductive Development of Plants

It is known that some oxylipins are required for regular plant growth and development, as such is the case of JA in maize, which is required for the formation of the male reproductive organ known as the tassel [25]. Few studies have examined the role of ketols in plant growth and development, but their involvement in these processes has been recently postulated based on a few lines of experimental evidence. Since *AOS* genes that contribute to 9/13-ketol production are expressed in the untreated roots of many plant species, it is likely that ketols not only have a role in defense against soil-borne pathogens but potentially in the regulation of root growth and differentiation [19,26,28,35,54]. Application of 10 uM 9,10-KODA to liquid culture enhanced root differentiation in senburi (*Swertia japonica* Makino), an herbaceous plant native to Eastern and Southern Asia that is known to produce antioxidants [54]. Interestingly, there seemed to be a threshold to the amount of 9,10-KODA that was beneficial for root differentiation, as 100 uM of 9,10-KODA inhibited root differentiation [54]. In the same study, dry weight and the number of adventitious roots increased in senburi grown in liquid culture supplemented with 10 uM of 9,10-KODA [54].

Ketols are likely involved in other plant physiological responses. For example, programmed cell death (PCD) occurs in Petunia flowers after pollination, during which a senescence-related gene *Psr2* was upregulated in floral tissue [55]. Although ketol production is not outright mentioned, the protein Psr2 was found to be closely related to LeAOS3, which has 9/13-AOS activity in tomatoes [26,55]. Psr2 was designated as a likely CYP74C enzyme and was also found to be localized to tonoplasts, contrary to the typical localization of 13-AOS in chloroplasts [55].

Since the early 20th century, several studies have identified putative floral-stimulating signals known as “florigens” [56]. However, not all these compounds act as universal florigens within flowering plant species; therefore, there is a continual search to identify florigen-acting compounds in plants [56]. One of the most interesting properties of ketols, notably 9,10-KODA, is their involvement in the induction of flowering in plants [57,58]. In the short-day plant *Pharbitis nil* Choisy, 9,10-KODA induction displayed a positive correlation with flower bud formation [58]. In this same species, the length of the dark cycle also influenced 9,10-KODA induction and flower bud formation, with a 16 h dark cycle being optimal for flower bud formation and 9,10-KODA induction [58]. However, ketol compounds may not act alone in flowering. Plants produce a variety of catecholamines that aid in the facilitation of various responses in plants, most notably to induce flowering and in response to oxidative, pathogen, and abiotic stresses [59,60]. The synthesis of catecholamines is akin to the mammalian synthesis of these compounds [59]. Surprisingly, 9,10-KODA has been shown to conjugate to norepinephrine in vitro to induce flowering in duckweed at a greater capacity than the individual compounds alone [61]. During various abiotic stresses (drought, heat, and osmotic stress), 9,10-KODA and norepinephrine were released into aquatic systems by duckweed [61]. However, it is unknown if such conjugation occurs in vivo.

**Figure 1 plants-12-02088-f001:**
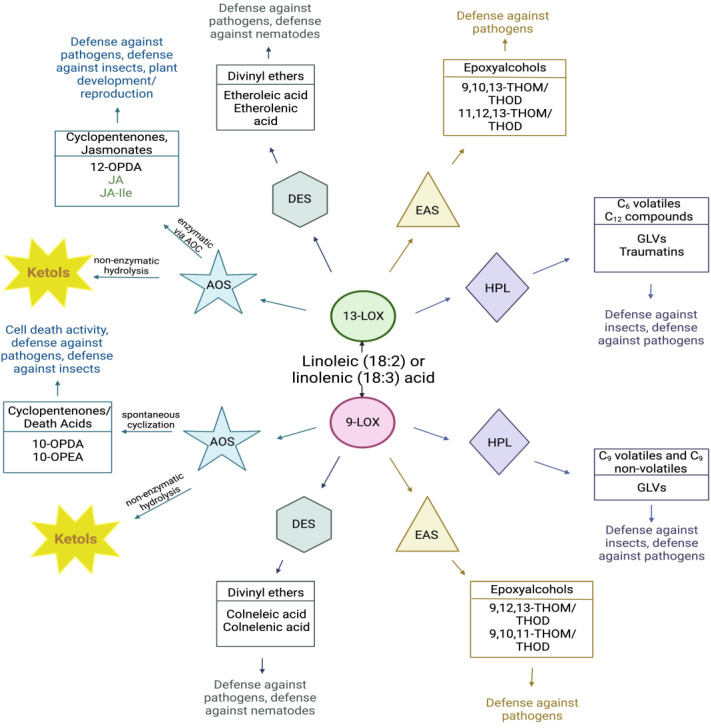
The CYP74 subfamily of enzymes gives rise to a diverse array of oxylipins (inspired by [24]). Linoleic (C18:2) or linolenic (C18:3) acids are acted upon by 13-LOXs or 9-LOXs to produce hydroperoxide fatty acids that are subsequently shuttled into CYP74 subbranches. These subbranches consist of divinyl ether synthase (DES), epoxyalcohol synthase (EAS), hydroperoxide lyase (HPL), and allene oxide synthase (AOS). The 13-AOS enzyme gives rise to jasmonic acid (JA) through subsequent enzymatic reactions. The 13-ketols are also produced via 13-AOS through non-enzymatic hydrolysis of an epoxide intermediate. The 9-AOS pathway gives rise to 9-ketols that, similar to 13-AOS, are produced through non-enzymatic hydrolysis. Other products of this pathway are 10-OPEA and 10-OPDA, which are known as “death acids”. These products can be synthesized via spontaneous cyclization; however, the 9-AOC pathway has been postulated to exist in plants [17,19]. The exact physiological function of many oxylipins is unknown, though they are implicated in playing important functions in plant growth, development, and stress responses. This figure was created with BioRender.com (accessed on 17 January 2023).

**Figure 2 plants-12-02088-f002:**
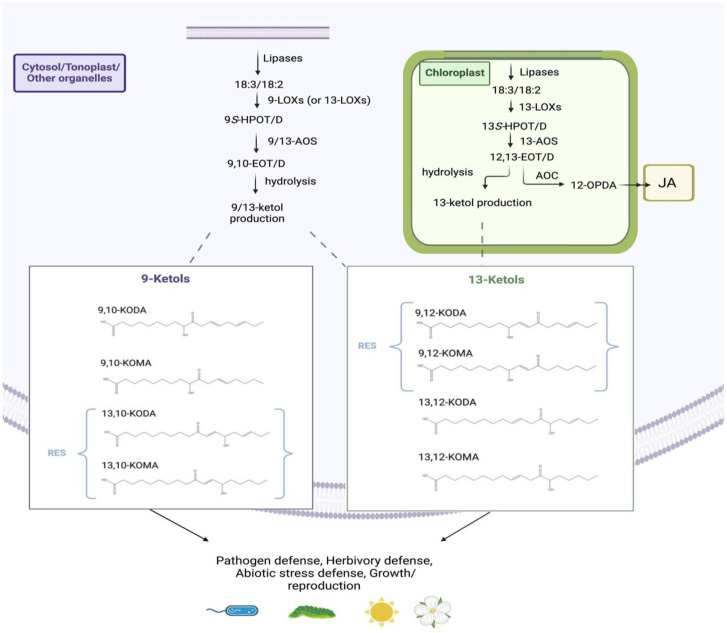
Biosynthesis pathways of ketols. The 13-AOS enzymes and production of 13-AOS ketol products are localized to the chloroplast, while the 9-AOS enzymes and production of 9-AOS ketols are localized to the cytosol or other organelles [6]. Ketols produced by these pathways through hydrolysis are as follows: 9-hydroxy-12-oxo-10(E),15(Z)-octadecadienoic acid (9,12-KODA), 9-hydroxy-12-oxo-10(E)-octadecenoic acid (9,12-KOMA), 13-hydroxy-12-oxo-9(Z),15(Z)-octadecadienoic acid (13,12-KODA), 13-hydroxy-12-oxo-9(Z)-octadecenoic acid (13,12-KOMA), 9-hydroxy-10-oxo-12(Z),15(Z)-octadecadienoic acid (9,10-KODA), 9-hydroxy-10-oxo-12(Z)-octadecadienoic acid (9,10-KOMA), 13-hydroxy-10-oxo-11(E),15(Z)-octadecadienoic acid (13,10-KODA), and 13-hydroxy-10-oxo-11(E)-octadecadienoic acid (13,10-KOMA). The 9,12-KODA, 9,12-KOMA, 13,10-KODA, and 13,10-KOMA are classified as reactive electrophilic species (RES). It should also be noted that, presently, a 9-AOC enzyme has yet to be identified. However, 10-OPEA and 10-OPDA can still be produced under spontaneous cyclization. This figure was created with BioRender.com (accessed on 9 December 2022).

**Figure 3 plants-12-02088-f003:**
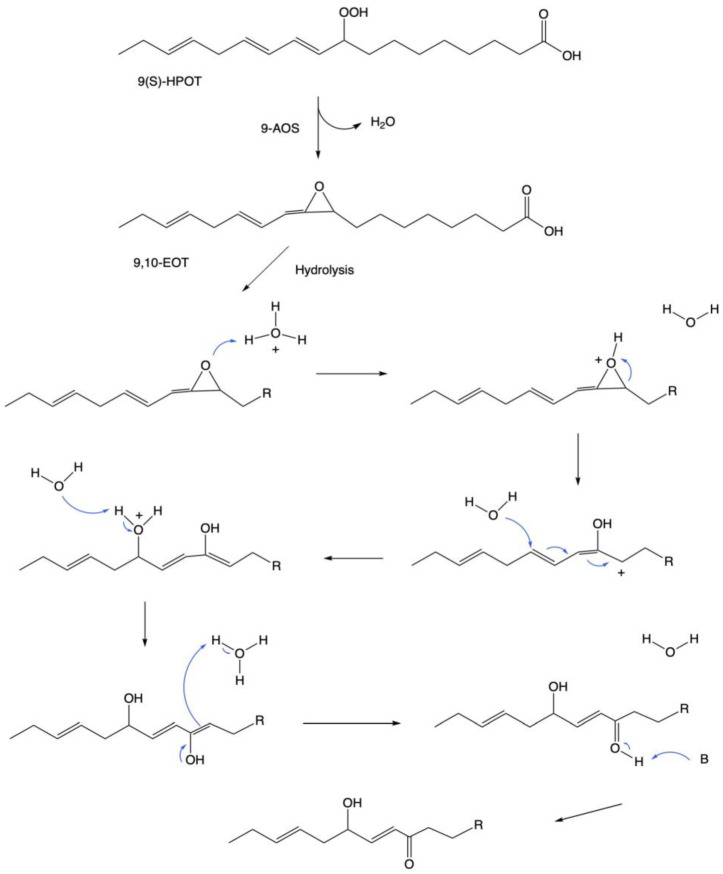
Proposed mechanism for a γ-9-ketol formed from a 9-hydroperoxide of linolenic acid. In this example, the beginning substrate is (9S,10E,12Z,15Z)-9-hydroperoxyoctadeca-10,12,15-trienoic acid, which is converted to an epoxide (9,10-Epoxyoctadecatrienoic acid) by 9-AOS. The following steps outline the hydrolysis reactions that lead to the formation of the final ketol product, 13,10-KODA. In the final step, a compound acting as a base (denoted as B) attracts the hydrogen in the carbonyl group. It should be noted that B could be water (H_2_O) or a basic site on an enzyme. Inspired by [15].

**Table 1 plants-12-02088-t001:** Citations for diverse ketol functions in physiological processes utilized in this review.

The Literature	Reference in Review	Plant Physiological Response Where Ketols Have a Role
Yan et al. 2012	[1]	Defense against insects
Wang et al. 2020	[22]	Defense against pathogens
He et al. 2020	[25]	Defense against insects
Itoh et al. 2002	[26]	Defense against pathogens
Stumpe et al. 2006	[28]	Defense against pathogens
Grechkin et al. 2000	[35]	Defense against pathogens
Endo et al. 2013	[40]	Defense against pathogens
Wang et al. 2016	[41]	Defense against pathogens
Wang et al. 2020	[42]	Defense against pathogens
Gorman et al. 2021	[43]	Defense against pathogens
Wang et al. 2021	[44]	Defense against pathogens
Chuang et al. 2014	[45]	Defense against insects
Tzin et al. 2015	[46]	Defense against insects
Tzin et al. 2017	[47]	Defense against insects
Koeduka et al. 2022	[48]	Defense against insects
Fitoussi et al. 2021	[49]	Defense against nematodes
Gao et al. 2008	[50]	Defense against nematodes
Gorina et al. 2022	[51]	Defense against abiotic stress
Haque et al. 2016	[52]	Defense against abiotic stress
Lõhelaid et al. 2015	[53]	Defense against abiotic stress
Kawakami et al. 2015	[54]	Plant growth/development
Xu et al. 2006	[55]	Plant growth/development
Corbesier et al. 2006	[56]	Plant growth/development
Yokoyama et al. 2000	[57]	Plant growth/development
Suzuki et al. 2003	[58]	Plant growth/development
Kulma and Szopa, 2006	[59]	Plant growth/development
Iriti et al. 2013	[60]	Plant growth/development
Yamaguchi et al. 2001	[61]	Plant growth/development

## 7. Conclusions

Plants produce a variety of ketols whose functions are vastly understudied. Many plant species, especially monocots, possess an AOS enzyme that is active towards 9- or 13-hydroperoxide substrates [6,19,24,26,28,33,34,35]. Although ketol compounds were previously thought to be the by-product of JA synthesis, recent exciting evidence suggests their involvement in the regulation of diverse physiological processes, including defense against biotic and abiotic stresses, flowering, and PCD processes (Table 1). Evolutionarily, the introduction of AOS genes is thought to have occurred before the diversification of land plants [14,34]. Even more interesting is the presence of dual-specific AOS enzymes being present in more monocot species of plants than dicot species. It is tempting to speculate if the reason for this diversification is due to the synthesis of ketols being important to the function and defense of these plant species or if this diversification occurred for other unknown reasons. Although 9,10-KODA, 9,12-KODA, and 9,12-KOMA are noted as important signaling molecules for SAR and ISR induction [22,40,42], it is unknown if other 9/13-AOS-derived ketols function similarly. Further research is needed to elucidate the role of specific molecular species of ketols to determine if they function similarly or in an additive manner and if they potentially act independently of other plant defense hormones, such as JA or SA, in various plant defense responses.

Although previous research has largely ignored the significance and importance of ketols in plant physiological and stress responses, available evidence suggests that ketols may be important regulators of these processes in plants. Future research using mutational, pharmacological, and overexpression approaches should attempt to elucidate the roles of diverse ketols in plants. Because some ketols display hormone-like signaling activities, it would be necessary to determine their mode of action, the extent by which they regulate transcriptome, proteome, and metabolome responses, and whether they serve as ligands for specific ketol receptors or post-translationally modify proteins as is the case for lipid-mediated protein acylation.

## Data Availability

No new data were created or analyzed in this study. Data sharing is not applicable to this article.

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
