# Peer review of "Ketols Emerge as Potent Oxylipin Signals Regulating Diverse Physiological Processes in Plants"

_plants, 2023, doi:10.3390/plants12112088_

Round 1

Reviewer 1 Report

This review presented in a high quality. Minor errors should be corrected all through the paper.

 The overall quality of English in this paper is good. Please correct the minor language errors in this paper.

Author Response

We thank the reviewer for the appreciation of the value of this review article.

Reviewer 2 Report

This is an interesting manuscript, but you may improve this article to publish in this journal. Otherwise, I have a lot of recommendations to increase the quality of your paper. Be careful with the writing and mistakes.

First, you must write the number of the lines in the review format. It seems that you do not have used the template of the journal properly.

You must delete the two points between “Correspondence” and “author”.

Just in the very beginning of the Abstract, just after the word “Abstract”, “Plants” is in bold which obviously is a mistake.

In the whole text when you write an acronym you must write in capitals the letters that you use to build it. This is a very common mistake. Please, fix it in the whole manuscript.

Just before “Introduction” you must write the number 1. Just after “Introduction” you must delete the two points.

As well, you must write a number for every section, for example, you must write “1.1. The lipoxygenase pathway: CYP74 branches”.

As well I there is no Results, only a big Introduction and Conclusions. It is not enough.

When you write the references from 1 to 4 there is a tiny mistake. You must follow the rules of the journal. You have written “[1, 2, 3, 4]” but the correct way is as follows: “[1–4]”. Please, fix this mistake in the whole article.

At the very end of the first paragraph you have written “[6, 7, 8, 9, 10, 11, 12].” However, you must follow the rules of this journal in order to publish your paper. The right way is as follows: “[6–12]”. Please, fix this tiny mistake in the whole paper.

You have written “Fig. 1” in the text or into brackets but the correct way to write it in this journal is writing it full length as follows: “Figure 1”. It seems that you do not have follow the instructions of this journal, please, read them again and follow all of them.

In the text, you have cited correctly Tzin et al. (2017) but once more, if you follow the rules of this journal you must write the number [49]. So, you must write the line as follows: “…study of Tzin et al. [49] which showed…”.

Finally, just because this is a botanical journal which name is “Plants” when you write a scientific name you must write its authors.

As well, you have written “C. imbricate” but you have forgotten the genus. Please, fix this mistake.

Moreover, for the references please, follow the template that you can download in the homepage of this journal.

Otherwise, the authors adequately developed the Introduction, presenting the problems but you must write explicitly the objectives of this paper.

There are no Results section as well.

There are no methods section. This is a big mistake.

There is no Discussion section comparing your results with other papers.

As well, you have write the author contributions, funding, data availability statement, acknowledgments and conflicts of interest.

Please fix all these big mistakes.

Just follow the rules of this journal.

The English is correct. I have detected only mistakes of the format of the journal.

Author Response

This is an interesting manuscript, but you may improve this article to publish in this journal. Otherwise, I have a lot of recommendations to increase the quality of your paper. Be careful with the writing and mistakes.

First, you must write the number of the lines in the review format. It seems that you do not have used the template of the journal properly.

Response: We agree with this reviewer and have re-formatted the manuscript to fit the template of the journal.

You must delete the two points between “Correspondence” and “author”.

Response: We agree with the reviewer, and reviewed grammatical/formatting errors such as this.

Just in the very beginning of the Abstract, just after the word “Abstract”, “Plants” is in bold which obviously is a mistake.

Response: We agree with the reviewer, and reviewed grammatical/formatting errors such as this.

In the whole text when you write an acronym you must write in capitals the letters that you use to build it. This is a very common mistake. Please, fix it in the whole manuscript.

Response: Corrected throughout the text.

Just before “Introduction” you must write the number 1. Just after “Introduction” you must delete the two points.  As well, you must write a number for every section, for example, you must write “1.1. The lipoxygenase pathway: CYP74 branches”.

Response: We agree with the reviewer. In order to better organize the flow of the review paper, we introduced each subsection of the introduction as ‘1.1, 1.2, etc.’

As well I there is no Results, only a big Introduction and Conclusions. It is not enough.

Response: As this is a review paper, we have not performed experiments that warrant a ‘Results’ section. We made sure that Abstract clearly states that it is a review paper.

When you write the references from 1 to 4 there is a tiny mistake. You must follow the rules of the journal. You have written “[1, 2, 3, 4]” but the correct way is as follows: “[1–4]”. Please, fix this mistake in the whole article.

Response: Corrected.

At the very end of the first paragraph you have written “[6, 7, 8, 9, 10, 11, 12].” However, you must follow the rules of this journal in order to publish your paper. The right way is as follows: “[6–12]”. Please, fix this tiny mistake in the whole paper.

Response: Corrected.

You have written “Fig. 1” in the text or into brackets but the correct way to write it in this journal is writing it full length as follows: “Figure 1”. It seems that you do not have follow the instructions of this journal, please, read them again and follow all of them.

Response: Corrected.

In the text, you have cited correctly Tzin et al. (2017) but once more, if you follow the rules of this journal you must write the number [49]. So, you must write the line as follows: “…study of Tzin et al. [49] which showed…”.

Response: Corrected.

Finally, just because this is a botanical journal which name is “Plants” when you write a scientific name you must write its authors. As well, you have written “C. imbricate” but you have forgotten the genus. Please, fix this mistake.

Response: Corrected.

Moreover, for the references please, follow the template that you can download in the homepage of this journal.

Response: We have followed the suggestion of this reviewer and have inserted our document into the journal template.

Otherwise, the authors adequately developed the Introduction, presenting the problems but you must write explicitly the objectives of this paper.

Response: The objectives of the paper have been outlined in the abstract and introduction. The goal of this review is to provide insight on the functions of ketols to highlight what is known about these oxylipins, and build a case for their importance to plant defense, growth, and reproduction.

There are no Results section as well. There are no methods section. This is a big mistake. There is no Discussion section comparing your results with other papers.

Response: Sorry for the confusion. This is a review paper.

As well, you have write the author contributions, funding, data availability statement, acknowledgments and conflicts of interest.

Response: We have added in author contributions, data availability, and conflicts of interest in addition to our acknowledgements and funding sections.

Please fix all these big mistakes.

Just follow the rules of this journal.

Reviewer 3 Report

The paper focuses on the role of Ketols as oxylipin signals in the regulation of plant physiology, including their involvement in plant growth, development, and response to stress. However, the article has not yet met the publication requirements. Some of the content and the structure of the paper need further improvement by the authors. My comments are as follows:

1. I think the author should give a better “Keywords”. 

2. In this review, the division of structure is considered unreasonable, as it contains only two parts, introduction, and conclusion. It is suggested that the author readjust the structure of the full text.

3. “Evolution of AOS enzymes and their ketol products” and “Biosynthesis and occurrence of ketols” sections, could the two parts be combined into one paragraph or divided in more detail?

4. It is suggested that the author enrich Figure 1. For example, add the known modifiable physiological functions of the various products.

5. Why is the biosynthetic pathway of ketones restricted to maize in Figure 2?

6. It is suggested that the authors update a table summarizing articles on ketones that respond to plant growth and development and in response to adversity stress.

7. The conclusion section does not mention useful information such as which physiological processes ketols are involved in and their mechanism of action.

I think the English is good

Author Response

The paper focuses on the role of Ketols as oxylipin signals in the regulation of plant physiology, including their involvement in plant growth, development, and response to stress. However, the article has not yet met the publication requirements. Some of the content and the structure of the paper need further improvement by the authors. My comments are as follows:

  1. I think the author should give a better “Keywords”. 

Response: We agree with the reviewer, and have given additional keywords. In the key words, we added “CYP74 enzymes”, “ketols”, and “conjugation to catecholamines” to offer a better overview of key words that fit the entire review

  1. In this review, the division of structure is considered unreasonable, as it contains only two parts, introduction, and conclusion. It is suggested that the author readjust the structure of the full text.

Response: Thanks for this suggestion. In order to better structure the paper, we divided the text into subheadings of ‘1.1, 1.2,’ etc.

  1. “Evolution of AOS enzymes and their ketol products” and “Biosynthesis and occurrence of ketols” sections, could the two parts be combined into one paragraph or divided in more detail?

Response: We agree with this suggestion and now combine the two sections under the chapter “Biosynthesis and occurrence of ketols”.

  1. It is suggested that the author enrich Figure 1. For example, add the known modifiable physiological functions of the various products.

Response: Additional functions of these compounds have been added into Figure 1. For additional information, we have included the citation of a few reviews that thoroughly outline the importance of these compounds.

  1. Why is the biosynthetic pathway of ketones restricted to maize in Figure 2?

Response: We have removed the word “maize” from Figure 2. This figure details the biosynthetic pathway of ketols for a variety of plant species, and not just maize.

  1. It is suggested that the authors update a table summarizing articles on ketones that respond to plant growth and development and in response to adversity stress.

Response: We agree with this reviewer, and produced a table that lists publications that discuss ketols and their releavnce to biotic/abiotic stresses, and growth/development.

  1. The conclusion section does not mention useful information such as which physiological processes ketols are involved in and their mechanism of action.

Response: We have used better phrasing to highlight the important physiological processes that ketols are implicated in. However, the cited references do not provide experimental details as the molecular mechanism behind their proposed physiological functions. Based on literature, ketols are only now emerging as an important group of oxylipins and this review intends to bring closer attention to these molecules as potential novel oxylipin signals worthy of further in-depth research.

Round 2

Reviewer 2 Report

This is an interesting and useful paper, but you may improve this article to publish in this journal. Otherwise, I have a lot of recommendations to increase the quality of your paper. Be careful with the writing and mistakes.

I have read all the corrections and you have done a good work.

But still there are other mistakes.

Line 18. Perfect. Now is much clear than before.

Lines 21-23. You must write in alphabetical order the keywords.

Line 25. I have realized that you have deleted “1. Introduction”. I cannot see this point. Please, write it again.

Line 28. Perfect. Now is much clear than before.

Line 39. Perfect. Now is much clear than before.

Line 157. Just because this is a botanical journal you must write all the authors of all the scientific names. Please, fix this mistake in the whole paper. So, you must look for all the scientific names in the manuscript and fix this mistake. For this reason, you must look for the author of Branchiostoma belcheri.

Line 164. Just because this is a botanical journal you must write all the authors of all the scientific names. Please, fix this mistake in the whole paper. So, you must look for all the scientific names in the manuscript and fix this mistake. For this reason, you must look for the author of Physcomitrium patens.

Line 176. Just because this is a botanical journal you must write the scientific name of tobacco.

Lines 178-179. You have repeated a mistake that I corrected in the previous version of this document. When you write an acronym, you must put in capitals the letters that you use to build the acronym so you must write “Systematic Acquired Resistance (SAR)”. This is more coherent now. Please, look for this mistake in the whole work and correct it.

Line 183. Just because this is a botanical journal you must write all the authors of all the scientific names. Please, fix this mistake in the whole paper. So, you must look for all the scientific names in the manuscript and fix this mistake. For this reason, you must look for the author of Glomerella cingulata.

Line 191. As I told you before in the previous revision you must follow the rules of this journal, so, you must delete the space between the two references. Therefore you must write the references as follows: “[22,44]”.

Line 196. Just because this is a botanical journal you must write all the authors of all the scientific names. Please, fix this mistake in the whole paper. So, you must look for all the scientific names in the manuscript and fix this mistake. For this reason, you must look for the author of Fusarium graminearum.

Line 196. You have repeated a mistake that I corrected in the previous version of this document. When you write an acronym, you must put in capitals the letters that you use to build the acronym so you must write “Gibberella Stalk Rot (GSR)”. This is more coherent now. Please, look for this mistake in the whole work and correct it.

Line 206. As I told you before in the previous revision you must follow the rules of this journal, so, you must delete the space between the references. Therefore, you must write the references as follows: “[1,47–49]”. Please, correct this very common mistake in the whole manuscript. Just follow the rules of this journal. As well you must use the long hyphen for consecutive references. Fix this mistake as well in your whole paper.

Line 261. As I told you before in the previous revision you must follow the rules of this journal, so, you must delete the space between the references. Therefore, you must write the references as follows: “[19,26,28,35,56]”. Please, correct this very common mistake in the whole manuscript. Just follow the rules of this journal.

Line 263. Just because this is a botanical journal you must write all the authors of all the scientific names. Please, fix this mistake in the whole paper. So, you must look for all the scientific names in the manuscript and fix this mistake. For this reason, you must look for the author of Swertia joponica. You must write the scientific name only the first time that you write the species in the text, so, in the line 267 you can write the plant without the authors.

Line 281. Just because this is a botanical journal you must write all the authors of all the scientific names. Please, fix this mistake in the whole paper. So, you must look for all the scientific names in the manuscript and fix this mistake. For this reason, you must look for the author of Pharbatis nil.

In the whole manuscript you have forgotten to justify the text on the right side of the lines, the previous review this aspect of the format was perfect but now you have done this mistake. Please, fix this blunder.

Please, fix all this mistakes and look for them in the rest of your work. I have realized that this article has improved a lot.

Otherwise, the authors adequately developed the Introduction, presenting the problems.

The authors are to be congratulated for the results obtained in this article.

Your English is understandable.

Author Response

Line 18. Perfect. Now is much clear than before.

Lines 21-23. You must write in alphabetical order the keywords.

Response: Corrected.

Line 25. I have realized that you have deleted “1. Introduction”. I cannot see this point. Please, write it again.

Response: We chose to number each section in numerical order, as there is no formal “introduction” for our review. Rather, the entire review is an introduction and in-depth review of ketol literature.

Line 28. Perfect. Now is much clear than before.

Line 39. Perfect. Now is much clear than before.

Line 157. Just because this is a botanical journal you must write all the authors of all the scientific names. Please, fix this mistake in the whole paper. So, you must look for all the scientific names in the manuscript and fix this mistake. For this reason, you must look for the author of Branchiostoma belcheri.

Response: Corrected.

Line 164. Just because this is a botanical journal you must write all the authors of all the scientific names. Please, fix this mistake in the whole paper. So, you must look for all the scientific names in the manuscript and fix this mistake. For this reason, you must look for the author of Physcomitrium patens.

Response: Corrected.

Line 176. Just because this is a botanical journal you must write the scientific name of tobacco.

Response: Corrected.

Lines 178-179. You have repeated a mistake that I corrected in the previous version of this document. When you write an acronym, you must put in capitals the letters that you use to build the acronym so you must write “Systematic Acquired Resistance (SAR)”. This is more coherent now. Please, look for this mistake in the whole work and correct it.

Response: Corrected.

Line 183. Just because this is a botanical journal you must write all the authors of all the scientific names. Please, fix this mistake in the whole paper. So, you must look for all the scientific names in the manuscript and fix this mistake. For this reason, you must look for the author of Glomerella cingulata.

Response: Corrected.

Line 191. As I told you before in the previous revision you must follow the rules of this journal, so, you must delete the space between the two references. Therefore you must write the references as follows: “[22,44]”.

Response: Corrected.

Line 196. Just because this is a botanical journal you must write all the authors of all the scientific names. Please, fix this mistake in the whole paper. So, you must look for all the scientific names in the manuscript and fix this mistake. For this reason, you must look for the author of Fusarium graminearum.

Response: Corrected.

Line 196. You have repeated a mistake that I corrected in the previous version of this document. When you write an acronym, you must put in capitals the letters that you use to build the acronym so you must write “Gibberella Stalk Rot (GSR)”. This is more coherent now. Please, look for this mistake in the whole work and correct it.

Response: Corrected.

Line 206. As I told you before in the previous revision you must follow the rules of this journal, so, you must delete the space between the references. Therefore, you must write the references as follows: “[1,47–49]”. Please, correct this very common mistake in the whole manuscript. Just follow the rules of this journal. As well you must use the long hyphen for consecutive references. Fix this mistake as well in your whole paper.

Response: Corrected.

Line 261. As I told you before in the previous revision you must follow the rules of this journal, so, you must delete the space between the references. Therefore, you must write the references as follows: “[19,26,28,35,56]”. Please, correct this very common mistake in the whole manuscript. Just follow the rules of this journal.

Response: Corrected.

Line 263. Just because this is a botanical journal you must write all the authors of all the scientific names. Please, fix this mistake in the whole paper. So, you must look for all the scientific names in the manuscript and fix this mistake. For this reason, you must look for the author of Swertia joponica. You must write the scientific name only the first time that you write the species in the text, so, in the line 267 you can write the plant without the authors.

Response: Corrected.

Line 281. Just because this is a botanical journal you must write all the authors of all the scientific names. Please, fix this mistake in the whole paper. So, you must look for all the scientific names in the manuscript and fix this mistake. For this reason, you must look for the author of Pharbatis nil.

Response: Corrected.

In the whole manuscript you have forgotten to justify the text on the right side of the lines, the previous review this aspect of the format was perfect but now you have done this mistake. Please, fix this blunder.

Response: Corrected.

Please, fix all this mistakes and look for them in the rest of your work. I have realized that this article has improved a lot.

Otherwise, the authors adequately developed the Introduction, presenting the problems.

The authors are to be congratulated for the results obtained in this article.

Response: Thank you so much for your thorough review and compliments of our work.

Reviewer 3 Report

I think the revised manuscript could be accepted for publication at its present form.

Author Response

I think the revised manuscript could be accepted for publication at its present form.

Response: Thank you for your thorough review of our work.